# The Role of the Endocannabinoid System in Binge Eating Disorder

**DOI:** 10.3390/ijms24119574

**Published:** 2023-05-31

**Authors:** Romain Bourdy, Katia Befort

**Affiliations:** Laboratoire de Neurosciences Cognitives et Adaptatives (LNCA), Université de Strasbourg, UMR7364, CNRS, 12 Rue Goethe, 67000 Strasbourg, France; bourdy@unistra.fr

**Keywords:** endocannabinoids, eating disorders, reward, diet-induced obesity, genetically modified mice

## Abstract

Eating disorders are multifactorial disorders that involve maladaptive feeding behaviors. Binge eating disorder (BED), the most prevalent of these in both men and women, is characterized by recurrent episodes of eating large amounts of food in a short period of time, with a subjective loss of control over eating behavior. BED modulates the brain reward circuit in humans and animal models, which involves the dynamic regulation of the dopamine circuitry. The endocannabinoid system plays a major role in the regulation of food intake, both centrally and in the periphery. Pharmacological approaches together with research using genetically modified animals have strongly highlighted a predominant role of the endocannabinoid system in feeding behaviors, with the specific modulation of addictive-like eating behaviors. The purpose of the present review is to summarize our current knowledge on the neurobiology of BED in humans and animal models and to highlight the specific role of the endocannabinoid system in the development and maintenance of BED. A proposed model for a better understanding of the underlying mechanisms involving the endocannabinoid system is discussed. Future research will be necessary to develop more specific treatment strategies to reduce BED symptoms.

## 1. The Endocannabinoid System and Food Intake

### 1.1. The Endocannabinoid System

Cannabis, specifically Δ-9-tetrahydrocannabinol (Δ^9^-THC) which is the psychoactive compound in the *Cannabis sativa* plant, is the most popular psychoactive substance in the world. Across the globe, approximately 209 million adults, including more than 22 million in Europe, reported using cannabis in 2020 [1,2]. Cannabis interacts with the endocannabinoid system (ECS), consisting of lipid neuromodulators (endocannabinoids) such as anandamide (AEA) and 2-arachidonoylglycerol (2-AG), their synthesis (NAPE-PLD/DAGLα) and degradation (FAAH/MAGL) enzymes, and two well-characterized receptors, the cannabinoid receptors CB1 and CB2. These receptors are coupled to G proteins (GPCR) of the Gi/o type. Other receptors, including the orphan GPR55 or PPAR, are also targets of cannabinoid compounds [3]. Ligands such as palmitoylethanolamide (PEA) and oleoylethanolamide (OEA) are examples of potential cannabinoid lipids interacting with PPARs. The CB1 receptor is the most predominantly expressed GPCR in the brain, in GABAergic and glutamatergic neurons [4], as well as in astrocytes [5]. Its role in the drug’s effect has already been widely described [6,7]. The CB2 receptor, first described as a peripheral receptor only, has also been identified in the hippocampus (HPC), striatum, and thalamus [8,9,10], and on the soma of dopaminergic (DA) neurons in the ventral tegmental area (VTA) [11]. Its functions in the brain remain largely unexplored. The ECS is clearly involved in numerous functions such as memory, emotional and motivational processes, as well as in pain management [12,13]. The ECS also modulates the rewarding effects, not only of cannabis, but of other drugs of abuse such as cocaine or opiates [14,15,16]. The role of this system in food intake has been widely described, by acting simultaneously through the central and peripheral mechanisms (gut, liver, muscle, and fat) [3,17].

### 1.2. The Endocannabinoid System Regulates Food Intake

Food intake is controlled by both peripheral physiological signals and brain reward and homeostatic systems (for a review, see [18]). These components interact via metabolic and nutritional signals that regulate homeostatic eating, and through reward signals that code the pleasurable aspects of food that drive hedonic intake. The homeostatic system is located in the mediobasal hypothalamus (HT), a region that includes the arcuate nucleus. Subpopulations of neurons in this area regulate feeding behaviors that are under the control of peripheral signals including ghrelin and leptin [19]. Hedonic eating is mediated by the reward system [20,21], which consists of the VTA DA projections to the nucleus accumbens (NAc), where reward and motivational signals are processed, as well as to the prefrontal cortex (PFC), a region involved in inhibitory control. Interestingly, among hypothalamic nuclei, the lateral hypothalamus (LH) regulates both the homeostatic and hedonic aspects of food intake, noticeably by its interaction with the reward system. Indeed, it is reciprocally connected to both the VTA and the NAc [18]. Moreover, VTA-projecting LH neurons encode reward-seeking actions and are involved in compulsive sucrose seeking [22]. In addition, LH neurons, in particular, glutamatergic ones, are involved in feeding suppression [23], and an obesogenic diet modulates glutamatergic reward-encoding properties [24].

Animal models have been developed to study feeding behaviors. Even if rodent studies cannot recapitulate all parameters of eating disorders, these have considerably improved our understanding of how differences in food access impacts these conditions [25]. Indeed, binge-eating behaviors in animal models informs our understanding of BED because binge eating is the primary characteristic of this disorder. Some of these paradigms will be described throughout this review to illustrate specific studies investigating the role of the ECS in food intake and more specifically in BED.

Although several components of the ECS may be involved, CB1 receptors appear crucial for the central and peripheral effects of cannabinoids on eating behavior [26]. CB1 receptors are widely expressed in reward-related brain regions [14], initially suggesting a role in the motivation to eat [26]. A well-known example of THC’s effect on eating behavior is marijuana-induced “munchies”. This phenomenon, recognized since ancient times and strongly conserved throughout evolution [27], corresponds to an intensified craving for standard and appetizing food that is commonly observed after cannabis consumption [28,29]. Pharmacological studies have further confirmed such a role. The systemic or local administration of endocannabinoids or CB1 agonists increase food intake in both humans and rodents [17]. For example, cannabis is well-known for activating hunger [28]. In rodent models, CB1 agonists such as Δ^9^-THC have orexigenic properties [30] and direct activation of hypothalamic structures by CB1 agonists increases food intake [31]. In addition, CB1 activation increases motivation to consume food by interacting with the reward system. For example, cannabinoid activation in the VTA, a key DA structure in the reward circuit, induces food intake of palatable food [32]. Infusions of the endogenous cannabinoid 2-AG directly into the NAc dose-dependently increase feeding in rats, an effect that is blocked by CB1 receptor antagonists [33]. This suggests that feeding controlled by the DA release in the NAc is regulated by the ECS. In addition, hypothalamic endocannabinoid levels are modulated in response to metabolic states with an increase during fasting and a decrease when animals are sated [3]. Hence, a functional link between endocannabinoids and DA activity has been reported and data suggest that the ECS regulates the incentive or motivational properties of food, thereby controlling food-seeking and -ingestive behavior [34]. Other endocannabinoids, such as OEA and PEA, also modulate eating behaviors by acting as a mediator of satiety [35]. OEA seems to interact with hedonic signaling, reducing the consumption of high caloric food [36]. In addition, the systemic administration of OEA dose-dependently prevents frustration-stress-induced feeding in female rats [37].

In opposition to CB1 activation, CB1 antagonists suppress food intake and weight gain in rodents [38,39,40,41]. Treatment with rimonabant, a selective CB1 antagonist, reduces excessive overeating and risk-taking behavior of compulsive eating in rats under a sweetened high-fat diet, as well as self-administration of a palatable beverage by decreasing palatable diet detection [34]. Mice given intermittent access to palatable food show an escalation of palatable food intake within the first hour of renewed access to this diet, an effect that is blocked by rimonabant [42]. Based on this evidence, the CB1 receptor blockade has been considered as a pharmacological tool to restore a normal endocannabinoid tone under pathologic conditions. Hence, the proposal that the CB1 antagonist, rimonabant, could be used to treat obesity in humans was an important step, although depressive and anxiogenic side effects rapidly ended this therapeutic approach [43,44].

Interestingly, several studies indicate that cannabidiol (CBD), a non-psychotic component of cannabis, may also regulate food intake, but the mechanisms by which this occurs is still poorly understood. This phytocannabinoid interacts with several molecular targets, including CB receptors, but whether CBD acts as an agonist or an allosteric modulator is unknown. Although some studies report a lack of CBD effects on food consumption, CBD appears to inhibit food intake [45,46,47] as well as responding for food or sweetened water in rats and monkeys [48,49]. A recent study has also shown that CBD inhibits oral sucrose self-administration through CB1 and CB2 mechanisms [50]. Nevertheless, more studies are needed to clarify how CBD acts in the ECS to modify feeding behavior.

From a circuitry perspective, CB1 receptors influence mesolimbic activity by inhibiting local GABA neurons or excitatory glutamatergic inputs to VTA-projecting NAc GABA neurons [51]. Recent electrophysiological studies have also shown an important role of endocannabinoid signaling in food intake, particularly in NAc D1-medium spiny neurons (MSN) projecting to the LH. Inhibition of these projections results in increased food intake. This synaptic inhibitory plasticity, associated with excessive food intake, is blocked/or induced by a CB1 antagonist/or agonist, respectively [52].

All the pharmacological approaches together with research using genetically modified animals (see below) have strongly highlighted a predominant role of the ECS in feeding behaviors. Whether this system is altered in BED needs to be clearly examined [53] for a better understanding of the adaptations of the hedonic balance in such situations.

## 2. Binge Eating Disorder

### 2.1. The Neurobiology of BED

BED is a psychiatric condition listed in the Diagnostic and Statistical Manual of Mental Disorders Version 5 (DSM-5) [54]. Among eating disorders, BED is the most prevalent in both genders with a high female/male ratio (6:4) [55]. It is characterized by uncontrollable episodes of eating within a short period of time, associated with loss of control. Recent studies examined associations between binge eating and addictive-like behaviors, highlighting gender differences [56].

Specific factors involved in the pathology and implementation of potential effective treatments have been recently reviewed for BED [57,58], and this strongly highlights the critical threat that they represent for public health.

Various neurotransmission systems including dopamine, opioid, endocannabinoid, and serotonergic systems are centrally involved in BED [59]. Moreover, homeostatic hormones involved in energy homeostasis such as ghrelin and leptin are often altered in BED, disturbing reward-system information processing [60,61]. Considering the extensive body of research examining the influence of the reward system on binge eating and other forms of maladaptive eating behaviors, we focused on the role of the DA reward system.

Preclinical, clinical, genetic, and neuroimaging data suggest that altered DA function in reward-related regions is an important contributor to BED. More specifically, reward imbalance (hypo-reactivity to conventionally rewarding stimuli and hyper-reward response to food-related stimuli) may contribute to the maintenance of eating disorders characterized by binge eating [62]. Indeed, repeated episodes of binge eating or exposure to highly palatable food may lead to a reduced responsivity of the reward system, decreasing motivation to engage in conventionally rewarding stimuli and necessitating the consumption of larger amounts of food or increased meal frequency to elicit the same response [63] (Figure 1A). Moreover, binge eating sensitizes the brain reward system to the anticipation of food, increasing its incentive salience and altering reinforcement learning (Figure 1B). This latter theory, the “incentive sensitization theory of binge eating”, is discussed in [64]. Impulsivity is a critical factor that contributes to the development of aberrant consumption patterns such as substance use disorders and eating disorders, playing a multifaceted role in the underlying etiology and pathogenesis of these conditions. Interestingly, evidence points to similar neurobiological mechanisms in compulsive eating and substance use disorders, the underlying alteration in the reward, motivation, and impulsivity-related processes [65]. The concept of “food addiction”, where people might be addicted to food by losing control over their aptitude to regulate food intake, is still debated [65,66].

### 2.2. Human Studies

Clinical and preclinical research have provided complementary knowledge using distinct types of techniques to investigate mechanisms underlying the neurobiology of BED. In humans, brain imaging technologies reveal that patients with BED show altered structural and functional changes in the reward system (for a recent detailed review, see [64]). Resting state functional connectivity studies reveal decreased fronto-striatal and inter-frontal connectivity in patients with BED. Importantly, decreased fronto-striatal connectivity is associated with a higher binge eating frequency [67]. In addition, fMRI studies reveal changes in reward responsiveness following the presentation of images representing palatable food. In patients reporting altered eating behaviors such as binge eating, these cues are responsible for the increased activity in the ACC, medial orbitofrontal cortex (OFC), and anterior insula, as well as an increase of the functional connectivity between the insula and medial OFC [68,69,70]. The latter is correlated with scores from the behavioral scale system, suggesting a relationship with the drive to acquire a reward [70]. Studies investigating brain responses to different kinds of rewards in patients suffering from BED show a lower responsivity to monetary rewards, but a higher responsivity to food rewards [71] in cortical subregions. Considering the DA system, patients with eating disorders that include binge episodes display a reduction in striatal DA transmission at rest with a lower pre-synaptic DA in the caudate nucleus, putamen, and NAc, suggesting a decreased basal DA activity [72,73]. On the other hand, they exhibit an abnormal initial response to reward, characterized by a significant increase in DA release in the caudate nucleus and putamen regions in response to taste stimuli [74]. This phenomenon, which is not observed in healthy controls, is positively associated with the frequency of binge-eating episodes and may contribute to reward system sensitization and increased cue salience. A pilot clinical trial recently brought promising results for the treatment of BED through NAc deep-brain stimulation [75]. In this paper, the authors managed to improve self-control of food intake and weight alterations by NAc deep-brain stimulation during food craving in two patients with obesity and BED. Binge episodes are also associated with altered habitual behavior, and more compulsive decision making [76,77]. Data from human studies reveal that BED is characterized by altered DA function, with increased food-reward sensitivity and impulsivity/compulsivity (for a recent review, see [78]). In this regard, cortico-striatal connectivity is involved in impulsivity, and these brain regions may therefore play a role in the development and/or maintenance of BED [59] (Figure 1C).

Other brain regions involved in cognitive processes may also contribute to binge-eating behavior. The HPC which is primarily associated with memory formation and retrieval is also involved in food-intake-related emotions and behaviors [79]. Some studies suggest that the HPC may be involved in regulating palatable-food-related craving [80]. A study reveals that patients with BED display reduced connectivity between the amygdala and PFC. Moreover, the amygdala is involved in palatable-food-associated impulsivity [81]. The amygdala could be involved in the tendency to overeat palatable food in response to negative emotional states, thus promoting palatable food approach behavior through striatal activation and PFC top-down inhibitory control suppression. One possible hypothesis is that the HPC and the amygdala play a role in the emotional regulation of hedonic food intake by modulating the activity of the striatal and cortical brain regions depending on past experiences and emotional states (Figure 1C). Future research is needed to better understand the role of the HPC and the amygdala in BED.

### 2.3. Animal Studies

Compared to clinical studies, examining binge eating in rodent models provides the benefit of evaluating different facets of eating patterns and nutrition, while avoiding the limitations of intricate cognitive, social, and cultural factors that typically accompany multifaceted eating disorders in humans. Preclinical research consistently highlights the pivotal significance of the DA reward system in BED. Palatable food consumption is sufficient to activate mesocorticolimbic brain regions [82,83] However, binge-eating-prone female rats display a greater activation (cFos/ΔFosB) of brain regions belonging to the reward system including the mPFC, NAc and VTA as compared to binge-resistant rats [84,85]. Moreover, deep-brain stimulation of the NAc shell reduces binge eating in mice through D2 receptors [86]. Animal studies seem to converge towards an alteration of DA signaling within the mPFC and NAc.

Binge eating induced by intermittent access to both standard chow and palatable food in rats is associated with increased DA release and turnover in the NAc during food access [87,88,89,90]. However, although alterations in DA levels are clearly linked to binge-eating behavior, whether the altered state leads to a hyper or hypodopaminergic state is still not clear and both states may co-exist depending on the pathological stage or interindividual differences (for a review, see [91]). These changes are accompanied by a disruption of DA signaling in the mesocorticolimbic system, through changes in DA-related gene expression and DA receptor density in the striatum [92,93]. Evidence from animal studies revealed that binge sucrose is related to increased D1 and reduced D2 receptor binding in the striatum [94,95], as well as reduced D2 receptor transcripts in the NAc [96] (Figure 1C). Moreover, a study comparing binge-eating-prone and -resistant rats show lower D1, D2, and D4 mRNA expression in reward-related brain regions, including the NAc and OFC, in binge-eating-prone male rats compared to control rats, with a negative correlation between D2 mRNA expression in the NAc and food consumed [97]. This suggests a link between NAc D2 receptors and the proneness to develop binge-eating behavior.

From a behavioral point of view, increased impulsivity is correlated with decreased striatal and midbrain D2 receptor levels and increased striatal DA release [98]. It is associated with greater bingeing behavior in both clinical and preclinical studies [99,100,101]. An important proportion of PFC glutamate neurons projecting to the NAc and the dorsal striatum (DS) modulate different behaviors, depending on the targeted brain region [102,103]. Importantly, dysfunction in this corticostriatal circuitry underlies impulsive behaviors [104,105]. In mice, optogenetic stimulation or optogenetic/pharmacogenetic inhibition of NAc-projecting PFC glutamate neurons reduces and increases fat intake in a binge procedure, respectively [106]. Interestingly, a recent study confirms the link between corticostriatal connectivity, impulsivity, and binge eating in rats [99]. Impulsive animals display significantly higher bingeing on fat compared to less impulsive ones, and chemogenetic activation of vmPFC-NAc shell neurons suppresses both motor impulsivity and binge-like intake of high fat food. These results point to the vmPFC-NAc pathway as a critical common neurocircuitry in the inhibitory control of both impulsivity and binge eating.

In addition to the PFC and striatum, other brain regions play a major role in binge eating (Figure 1C). The anterior insula encodes the organoleptic components of food (taste, flavor, and oral texture) and responds to its rewarding properties [107]. Its projections to the ventromedial putamen are thought to translate the sensory-interoceptive hedonic aspects of feeding into motivated motor behaviors for palatable food [108,109]. In female rats, the optogenetic inhibition of NAc neurons projecting to the anterior insula reduces the progressive ratio (PR) responding selectively in a high-responder subset of rats with intermittent access to chocolate-flavored pellets [61]. This pathway therefore seems to be involved in binge-eating-associated compulsive-like behaviors. As in humans, a role of the HPC is also described in BED animal models [110,111,112].

### 2.4. A Proposed Model for Altered Dopamine Activity and Reward Circuit Connectivity in BED

Altogether, data from patients suffering from BED and studies from BED animal models show alterations in DA transmission within the mesocorticolimbic system that may contribute to the altered reward sensitivity, motivation, and increased impulsivity observed in BED. Although the direction of such changes is not clear, results concerning DA tone point toward a hypodopaminergic resting state along with DA hyper-responsivity in response to palatable food or food-associated cues. In BED animal models, repeated binge episodes are responsible for an overactivated mesocorticolimbic system that could account for hedonic desensitization, leading to increased seeking for palatable food, frequency of bingeing episodes, and higher levels of food consumption. This would contribute to the maintenance of binge eating, engaging individuals in a vicious cycle of maladaptive food intake (Figure 1A). On the other hand, perpetuation of binge eating could reflect increased reward sensitivity to palatable-food-related cues, enhancing reinforcement learning for food, further leading to impulsive behavior (Figure 1B).

In terms of neuronal circuitry, preclinical studies inform our understanding of the networks involved in binge eating (see Figure 1C). Animal studies implicate the PFC and striatal D2 receptors in binge eating of palatable foods. The decrease in D2 receptors in the striatum, and more specifically in the dorsolateral striatum, leads to compulsive drug and food consumption [113]. Importantly, numerous rodent studies converge toward a link between binge eating and decreased striatal D2 receptors. In the striatal complex, D1 receptors, involved in reward, are predominantly expressed by MSNs in the direct pathway, whereas D2 receptors, involved in behavioral flexibility, are mainly expressed by neurons in the indirect pathway. Interestingly, impulsivity is correlated with ventral striatal levels of D2 receptors in rodents [98], and hypofunction of the indirect pathway is linked to perseverative and compulsive behaviors as observed in models of drugs of abuse. An imbalance in signaling between the direct and indirect pathways associated with decreased striatal D2 receptor expression and higher or unchanged D1 receptor expression may contribute to compulsive food intake observed in binge eating (for a review, see [59]). Animal studies showing the involvement of the PFC-NAc pathway and reduced PFC-NAc D2 pathway activity in binge-eating behavior and impulsivity are in line with this hypothesis [99,106]. Moreover, increased activity in both cortical and subcortical reward subregions and alterations in corticostriatal connectivity observed in patients with BED and animal models of BED suggest that binge-eating proneness may be the result of dysregulated executive control of the PFC over striatal regions rather than a unique alteration in reward perception and responsivity. Indeed, bingeing may be a consequence of an impaired top-down executive control by the PFC over the basal ganglia, thus affecting the control of reward responsivity and food-related impulsivity. Additionally, other brain regions such as the insula, HPC, and amygdala are also involved in the proprioceptive and emotional regulation of palatable food intake and may display altered function in binge eating (Figure 1C).

## 3. Binge Eating Disorder and the Endocannabinoid System

In addition to its involvement in appetite regulation and reward-based eating, several investigations indicate that the ECS is involved in many aspects of eating disorders and obesity [3,26,114]. Interestingly, studies in patients with psychiatric conditions have revealed relationships between ECS gene polymorphisms and their pathological state [114]. The most studied ECS gene in this context is *FAAH*, with polymorphisms described for patients with obesity and BED [115]. Similar polymorphisms have been described in drug addiction, supporting the idea of shared mechanisms involving the ECS across these pathological conditions [114]. Available literature results from various measurement methods including pharmacological, molecular, and biochemical assays, but the specific details are nevertheless difficult to disentangle in that several studies have yielded conflicting results, and this will be discussed in this section.

### 3.1. Endocannabinoid Levels in BED

Data from peripheral clinical and preclinical studies suggest a strong link between binge eating and endocannabinoid levels. Indeed, blood levels of AEA, but not 2-AG, are increased in BED drug-free women, an effect that is not observed in people with bulimia nervosa [116]. Interestingly, changes in peripheral levels of AEA in patients with obesity and BED are positively correlated with the “urge to eat” sensation, the amount of favorite food consumed, and the subjective pleasantness associated with food intake [117]. Another clinical study in women with obesity (or who are overweight) and BED confirms this link between BED and elevated plasmatic endocannabinoid levels with higher levels of both AEA and 2-AG compared to women without BED, and a predictive value of AEA blood levels for the disorder [118]. Finally, similar results are described in recent preclinical studies showing increased plasma levels of AEA and 2-AG in bingeing rodents [119,120] (Figure 2A).

At the central level (Figure 2B), animal studies provide the opportunity to follow the consequences of binge eating on endocannabinoid levels in brain regions involved in homeostatic and hedonic feeding regulation. While peripheral endocannabinoid levels tend to increase in response to binge eating, central endocannabinoid levels exhibit bidirectional regulation depending on the brain region studied, palatable food access schedule, diet composition, and gender. In the PFC, AEA levels are decreased in female rats bingeing on fat in a low-restriction access (LR; 2 h/day) [35] and increased in male rats bingeing on sucrose [121]. In striatal regions, one study reports a decrease in AEA in the DS and NAc following high- and low-restriction schedules, respectively, in female rats bingeing on fat. This suggests a differential regulation of AEA tone depending on the length of palatable food restriction. Additionally, bingeing on fat also reduces AEA levels in the HPC following an LR; in contrast, AEA decreases in the amygdala following fat bingeing independently of access schedule [35]. Interestingly, a frustration stress paradigm produces an opposite pattern of regulation in female rats bingeing on sweetened fat food, displaying increased AEA levels in the HT [120]. As in the HT, food deprivation increases endocannabinoid levels whereas food consumption decreases these in the limbic forebrain, including the NAc [33]. Alterations in 2-AG levels in animal models of binge eating are found to be restricted to the HPC, with female rats showing increased 2-AG levels after fat bingeing [35], and male rats showing decreases following sucrose bingeing [121]. Interestingly, decreased OEA levels have also been reported in the HT of female rats following fat bingeing [35], indicating the global regulation of lipids interacting with the ECS.

### 3.2. Catabolic and Anabolic Cannabinoid Enzymes in BED

#### 3.2.1. Transcriptional Regulations of the Cannabinoid Enzymes in BED

In addition to altered brain levels of endocannabinoids, some preclinical studies report alterations in endocannabinoid synthesis and degradation enzyme transcript levels in animal models of binge eating (Figure 2B). Thus, DAGL mRNAs levels are decreased in the NAc of male rats following a binge sucrose procedure [121] and are increased in the HT of female rats following a sweetened high-fat binge protocol [120]. In the latter study, FAAH mRNA levels are also increased in the HT, and associated with a decreased DNA methylation at the FAAH promoter. This FAAH mRNA changes could reflect a compensatory mechanism for the degradation of abnormally elevated local AEA levels.

#### 3.2.2. Genetic Animal Models Targeting the Endocannabinoid Enzymes to Study Food Intake and Addictive-like Eating Behaviors

Several genetic mouse models have been generated to probe ECS functions [16,122], including responses to palatable food. Table 1 summarizes the findings from this work, showing food intake and body weight changes of knockout (KO) mice for different components of the ECS, including the endocannabinoid hydrolyzing enzymes (FAAH and MAGL), the synthesis enzyme (DAGL), or double-mutant combining receptors and/or enzymes.

Considering the hydrolyzing enzyme MGL, there are some discrepancies in studies reporting the full deletion of the gene in mice, in terms of body weight and food intake. A first in vivo genetic evidence that this serine hydrolase contributes to the maintenance of 2-AG levels has been provided and the authors show that full-MGLKO mice exhibit lower body weight both in males and females [123]. Other reports indicate lean phenotypes at baseline or following LFD or HFD in both sexes [124], or no change in weight following a chow or HFD regimen [125]. Total food intake is similar between MGLKO and WT mice, but when using a food monitoring system allowing more precise measures, an increase in LFD is observed after 10 weeks, only in the light-feeding case [124]. In another study, no differences are observed in body weight and food intake when mutants are fed with regular chow, whereas, when HFD-fed, the MGLKO mice show reduced weight and obesity resistance [126]. When HFD exposure is prolonged over 24 weeks, even though the mutant mice exhibit a preference for HFD over normal chow, they finally become obese. These controversial observations may originate from the diverse functional roles of MGL in vivo. A clear perturbation of whole-body energy metabolism in mice overexpressing MGL specifically in the small intestine (iMGL mice) is observed, suggesting a role of the ECS component in the intestine in food intake and energy balance [127].

**Table 1 ijms-24-09574-t001:** Genetically modified mice of the ECS components in food intake and addictive-like behaviors. Several mutant mice are listed depending on the ECS gene that has been targeted. Models use classical chow exposure (Chow) and more sophisticated paradigms, including various types of palatable food and distinct schedules of exposure. For example, simple food-exposure protocols with unlimited access to high-fat diet (HFD) are proposed as a diet-induced obesity model, often used to mimic excessive intake of high-calorie diets as observed in humans. These are often compared to low-fat diet (LFD) exposure. Several paradigms would alternate fasting and refeeding episodes (fasting–refeeding) to investigate feeding behaviors. Other models try to mimic BED with high intake of palatable food (fat, sucrose, and chocolate) in a short time period, using intermittent access to palatable food. Complex operant paradigms have also been recently designed to examine addictive-like eating behaviors, with fixed ratio (FR) and progressive ratio (PR) schedules of reinforcement. In all these models, weight and behavioral responses towards food are usually reported, even though there is not always a correlation between weight changes and feeding behaviors. # correspond to a rescue construct (see text).

Target		Reference		Regimen	Body Weight
CB1 Full KO		Ledent, 1999	[128]	Chow	Similar
		Zimmer, 1999	[129]	Chow	Similar
		Varvel, 2002	[130]	Chow	Reduced
		Di Marzo, 2001	[131]	Chow	
				Fasting–refeeding	
		Cota, 2003	[132]	Chow	Reduced
		Song, 2011	[133]	Chow	Similar
		Bellochio, 2010, 2013	[134,135]	Fasting–refeeding	
		Massa, 2010	[136]	Chow	Reduced
				HFD	Reduced
		Liu, 2012	[137]	Chow	Similar
				HFD 14 wk	Reduced
		Osei-Hyiaman, 2005, 2008	[138,139]	Chow	Reduced
				HFD 14 wk	Reduced
		Yoshida, 2019	[126]	HFD	Reduced
		Soria-Gomez, 2014 #	[140]	Fasting–refeeding	
		Ravinet Trillou, 2003	[141]	HFD 6 wk	Reduced
		Ravinet Trillou, 2004	[142]	Chow	Reduced
				Free-choice chow + HFD	Reduced
		Bura, 2010	[143]	Free-choice HFD + sweet solution	Reduced
		Brommage, 2008	[144]	HFD	Reduced
		Powell, 2015	[145]	Chow	Reduced
				HFD	Reduced
		Quarta, 2010	[146]	Chow 12 wk	Reduced
				HFD 12 wk	Reduced
		Poncelet, 2003	[147]	Chow	
				Two-bottle choice sucrose 5%	
		Sanchis Segura, 2004	[148]	Two-bottle choice sucrose 5%	
				Operant sucrose 5%	
CB1 Full KO		Ward, 2005	[149]	Operant sweet food	
				Operant corn oil	
		Guegan, 2013	[150]	Operant chow/fasted	
				Operant highly palatable food	
				Operant highly isocaloric palatable food	
		Mancino, 2015	[151]	Operant chocolate pellet	Reduced
		Bi, 2020	[50]	Operant sucrose 5%	
CB1 cKO nervous system	CamK-CB1KO	Quarta, 2010	[146]	Chow 12 wk	Reduced
				Fasting–refeeding	
				HFD 12 wk	Reduced
		Bellochio, 2013	[135]	Fasting–refeeding	
	Glu-CB1KO	Lafenetre, 2009	[152]	Repeated exposure to novel palatable food	
		Bellochio, 2010, 2013	[134,135]	Fasting–refeeding	
				Exposure to palatable food in fed animals	Similar
		Domingo-Rodriguez, 2020	[153]	Operant food addiction	

		Ruiz de Azu,a 2021	[154]	LFD	Similar
				HFD 12 wk	Reduced
				Operant chocolate pellet following LFD/HFD	Reduced
				Free-choice LFD + HFD	Reduced
		Soria-Gomez, 2014	[140]	Fasting–refeeding	
	Glu CB1 RS	Soria-Gomez, 2014	[140]		
	GABA-CB1KO	Lafenetre, 2009	[152]	Repeated exposure to novel palatable food	

CB1 cKO nervous system		Bellochio, 2010, 2013	[134,135]	Fasting-refeeding	
				Exposure to palatable food in fed animals	Similar
		Massa, 210	[136]	Chow	Similar
				HFD	Similar for 10 wk, then reduced
	Glu/GABA CB1KO	Bellochio, 2010	[134]	Fasting–refeeding	
	TPH2-CB1KO	Bellochio, 2013	[135]	Fasting–refeeding	
	SF1-CB1KO	Bellochio 2013	[135]	Fasting–refeeding	
		Cardinal, 2014	[155]	Chow	Similar
				HFD 8 wk	Increased
	Sim-CB1KO	Bellochio, 2013	[135]	Fasting–refeeding	
		Cardinal, 2015	[156]	chow	Similar
				HFD 12 wk	Reduced
	Hyp-CB1KO	Cardinal, 2012	[157]	Chow	Reduced
				Fasting–refeeding	Similar
	Thy1-CB1KO	Pang, 2011	[158]	LFD	Reduced
				HFD 12 wk	Reduced
CB1 cKO periphery	htgCB1KO	Liu, 2012	[137]	Chow	Similar
				HFD 14 wk	Reduced

	LCB1KO	Osei-Hyiaman, 2008	[139]	chow	Similar
				HFD 14 wk	Similar
	intCB1KO	Avalos, 2020	[159]	Western diet preference	
CB2KO		Agudo, 2010	[160]	Chow	Increased
				HFD 8 wk	Reduced
		Deveaux, 2009	[161]	HFD 15 wk	Reduced
		Flake, 2012	[162]	Chow	Increased
		Pradier, 2015	[163]	Chow	Increased
		Bi, 2020	[50]	Operant sucrose 5%	
		Garcia Blanco 2023	[164]	Operant chocolate pellet	Similar
CB2xP		Romero-Zerbo, 2012	[165]	Fasting–refeeding	Reduced
		Garcia Blanco, 2023	[164]	Operant chocolate pellet	Reduced
Enzymes	DAGLKO	Gao, 2010	[166]	Chow	Reduced
		Powell, 2015	[145]	Chow	Reduced
	NAPE Pld	Powell, 2015	[145]	Chow	Similar
	MAGLKO	Chanda, 2010	[123]	Chow	Reduced
		Taschler, 2011	[125]	Chow	Similar
				HFD 10 wk	Similar
				Fasting–refeeding	
		Douglass, 2015	[124]	LFD	Reduced
				HFD	Reduced
	iMGL	Chon, 2012	[127]	Chow	Similar
				HFD 3 wk	Increased
	FAAH	Cravatt, 2001	[167]	Chow	Similar
		Tourino, 2010	[168]	Chow	Increased
				HFD 12 wk	Increased
				Operant chow	
				Operant chocolate	
				Operant fat	

In a high-throughput phenotypic screen study of over 3000 mutants, DAGLaphaKO and CB1KO mice show a similar lean phenotype on a chow regimen, whereas DAGLbetaKO and NAPEpldKO mice have no differences in weight, compared to WT mice [145]. Similar findings are reported, with decreased body weight in male and female DAGLalphaKO mice by 23 and 12%, respectively, and no significant changes in DAGLbetaKO mice [166]. Under an HFD diet, similar results are described for DAGLaphaKO and CB1KO mice, with 21% and 18% decreased weight, and 47 and 45% decreased body fat, respectively, with a significantly reduced food intake [145]. Together, these results confirm data for CB1KO mice (see below), and indicate that DAGLalpha alone provides the endocannabinoid 2-AG that serves as the endocannabinoid signal.

Considering the enzyme degrading AEA, FAAH, an initial study investigating FAAHKO shows no differences in weight or food intake in these mutants [167]. In contrast, another study reveals increased body weight in mutants in comparison with WT mice, both when exposed to chow or to HFD [168], which represents an opposite phenotype to CB1KO mice (see below). Interestingly, total daily food intake is similar between both genotypes, but differences occur during the dark phase (increase) or light phase (decrease) for high-fat food intake, suggesting that AEA plays an important role in the modulation of feeding in the light and dark periods. FAAHKO mice trained for different types of food (chow, fat pellets, or chocolate pellets) in an operant paradigm exhibit distinct phenotypes. Data reveal that the responding for standard or chocolate pellets is higher in FAAHKO, whereas no genotype difference is observed for fat pellets, both in fixed ratio 1 (FR1) and FR5, suggesting an enhancement of the reinforcing properties of these types of food. Motivation, evaluated with a PR schedule of reinforcement, is higher in FAAHKO for the three types of food, compared to WT mice, despite no differences in total food intake. These results indicate that FAAH plays an important role in the control of energy balance, probably by food-intake-independent mechanisms [168].

Altogether, these mutant data highlight a crucial role for endocannabinoid enzymes in the control of energy balance, and, together with transcriptional regulations, point to a specific role of both anabolic and catabolic enzymes in binge-like behavior for maintaining the dynamic of the endocannabinoid tone.

### 3.3. Cannabinoid Receptors in BED

#### 3.3.1. Transcriptional Regulations of Cannabinoid Receptors

Changes in endocannabinoid signaling, largely mediated by CB1 receptors, are linked to the development of binge eating disorder. Several studies report regulation of the brain ECS in preclinical and clinical conditions associated with various eating disorders, as well as in cases of food addiction and obesity [120] (Figure 2B). In these models, CB1 receptor mRNA is decreased in reward-related brain regions including the PFC [169] and the NAc [170] in animals bingeing on fat. Importantly, this decrease is associated with a decrease in CB1 receptor density within the same brain regions following fat bingeing compared to animals with ad libitum feeding schedules [35,169]. On the other hand, one study reports an opposite regulation, with an increase in CB1 mRNA in the NAc and HT following sucrose bingeing in male rats [121], suggesting a differential involvement of the CB1 receptor in the appetite for dietary components (fat vs. sucrose) in reward brain regions. Indeed, the biological mechanisms underlying fat and sucrose bingeing appear distinct [171]. Due to the lack of data, future studies on the effects of palatable food type on ECS gene expression are needed to address this question.

#### 3.3.2. Genetic Animal Models Targeting the Cannabinoid Receptors to Study Food Intake and Addictive-like Eating Behaviors

The most studied component of the ECS involvement in eating behaviors using genetically modified mice is the CB1 receptor. As CB1 receptors are mostly described in the brain, it was first proposed that food intake modulation was controlled through central sites, but its expression in the periphery also highlighted potential other mechanisms of energy-balance control. To investigate the anatomical sites involved in these processes, several approaches have been used, with a full KO of the gene, deletion in specific structures (conditional KO), or CB1 receptor expression rescue in specific brain regions in a deficient background. The reported data concern mainly body weight, food intake, and motivation or seeking behavior towards regular or palatable food (see details in Table 1).

Full knockout of CB1 receptors

Some studies, including the initial ones describing the first CB1KO mice, reported a normal body weight in these mutant mice [128,129,133,137]. In contrast, several other studies noted a significantly reduced body weight of the CB1KO mice [130,132,136,138,139,146,151]. These mutant mice have been followed on distinct feeding paradigms to investigate the involvement of CB1 receptors on fat and/or sweet intake. On an HFD, CB1KO mice appear resistant to the obesogenic effects of the diet [126,136,137,138,139,141,142,146], on which wild-type (WT) mice become obese. In a high-throughput screening study investigating the lean or obese phenotype of deficient mice, CB1KO mice are clearly identified as lean on an HFD [144]. On a combined free-choice chow and HFD, CB1KO choose the highly palatable diet, but reduce their intake and therefore do not develop an obese phenotype [142]. Interestingly, CB1KO on HFD do not show changes in their metabolic and hormonal profile and do not develop fatty liver despite having a caloric intake similar to that of WT mice on the HFD [138], suggesting resistance to obesity.

Food intake of deficient CB1 mice appear more variable depending on the study or on the pattern of access. Food intake with unlimited access is similar to WT [131,138,142,146,147]. Reduced calorie intake is observed in a fasting–refeeding paradigm with standard food, known to activate the ECS, using CB1KO [131,134,135], or using a stop-CB1KO, which is equivalent to a KO animal [140]. Similar reductions are observed in CB1KO with full access to both a standard [132,136] or HFD [136] diet. A free-choice feeding paradigm, with both an HFD and sweet solution, result in an HFD feeding with a progressive gain of body weight [143]. No differences between genotypes are revealed under basal conditions in the preference for HFD, whereas CB1KO mice show reduced body weight and HFD intake. Interestingly, in the same paradigm, mutant mice show a lower saccharin preference. Similar results are observed in an operant paradigm for either sweet food or corn oil, where mutant mice take longer to acquire operant responding for the sweet food (FR1 schedule of reinforcement) and obtained a reduced point break (on a PR schedule of reinforcement), indicating a lower motivation, whereas no changes compared to WT are observed for the operant responding for fat [149]. Reduced intake of sucrose in CB1KO mice compared to WT is also observed in a two-bottle choice procedure [147,148] and under operant conditions [50,148]. The latter study also shows a decreased consumption of saccharin in CB1KO mice, suggesting that this effect was independent of the caloric value of the sweet solution [148]. Interestingly, a decrease in intake and operant responding is observed with standard, highly isocaloric (as standard) palatable food and highly palatable food in fasted CB1KO animals, whereas it is only observed for highly palatable isocaloric food in non-fasted mutant mice [150]. These data indicate that highly palatable isocaloric food strongly promotes operant behavior and enhances motivation for food through a CB1-receptor-dependent mechanism. In addition, CB1KO mice trained with chocolate-flavored pellets significantly reduce operant active responses during FR1 and FR5 compared with WT, highlighting reduced operant seeking behavior in CB1KO [151].

All together, these results demonstrate that the CB1 receptor is a key component in the development of diet-induced obesity, and also highlight its preferential role in the reinforcing effect of sweet as compared to fat food. 

Conditional CB1 knockout mice in the central nervous system

As CB1 receptors are expressed in distinct neuronal populations, they may differentially regulate feeding behavior. Some studies have investigated the effect of the partial neuronal deletion of CB1 on food intake. Transgenic mice expressing an artificial microRNA under the control of the neuronal Thy1.2 promoter allowed the knockdown of the CB1 receptor in the central nervous system [158]. These mutant mice display reduced body weight and feeding efficiency when fed chow or HFD. This phenotype differs from the full CB1KO, an observation that may account from CB1 receptors expressed in the neurons of the peripheral nervous system or in unidentified non-neuronal cells. Several studies using the Cre/loxP system have investigated the role of specific CB1 receptors in the CNS in body weight and food intake behaviors (Table 1). The first studies compare cortical glutamatergic and GABAergic transmissions controlled by CB1 receptors and reveal unexpected opposing brain functions of the CB1 receptors in the regulation of food intake [134,135,152]. Normal body weight is described for these mutant mice fed on normal chow. As with the full CB1KO, reduced food intake is observed in Glu-CB1KO, whereas the opposite behavior with hyperphagia is observed in GABA-CB1KO, in a fasting–refeeding paradigm [134,135]. A similar finding is obtained following exposure to palatable food in fed animals [134]. Interestingly, the double-mutant GABA-Glu-CB1KO has a similar stimulated-food-intake phenotype compared to WT animals [134], suggesting that the two mutations compensate for each other. Another study using a fasting–refeeding paradigm shows similar results for Glu-CB1KO, a phenotype that is fully restored in a rescue approach, where the CB1 receptors were only expressed in Glu neurons (on a KO background in a stopCB1 mouse line), indicating that CB1 in Glu neurons are sufficient to provide a normal fasting-induced food intake [140]. When mice are subjected to repeated exposure to novel palatable food, Glu-CB1KO eat less than WT, and take more time to approach the pellets, suggesting increased behavioral inhibition. Glutamatergic receptors may therefore be involved in the pleasure of eating. In similar conditions, GABA-CB1 mice eat more palatable pellets from the first session, a hyperphagic behavior that is maintained throughout the experiment, indicating a role for these receptors in safety behavior and impulsivity [152].

Glu-CB1KO mice show a similar body weight as WT when exposed to LFD, as previously described [135], whereas weight gain and food intake are reduced on an HFD [154]. A similar phenotype is observed in an operant paradigm with chocolate pellets or in a free-choice LFD/HFD regimen, even though the mutant mice show a preference for HFD, as did WT mice [154]. Considering the role of GABA-CB1 receptors in obesity development, mutant mice present a similar body weight as WT mice on regular food, whereas they show reduced weight on a late HFD regimen, with a reduced amount of visceral fat, suggesting a resistance to obesity in a late phase of emergence of the disease, despite similar caloric intake [136]. Interestingly, in a preclinical model to study resilience and vulnerability to develop food-addiction-like behavior, mice lacking CB1 receptors in glutamatergic neurons are characterized by less perseverance, reduced motivation, and decreased compulsivity for highly palatable food compared with WT mice [153]. Furthermore, chemogenetic inhibition of neuronal activity in the PFC-NAc pathway induces compulsive food seeking. These Glu-CB1KO mice do not fully mitigate the development of food addiction in this paradigm, but significantly promote a resilient phenotype.

Altogether, these results indicate that, by altering palatable food intake depending on the pattern of access and the type of diet, CB1 deletion in neuronal subpopulations is specifically altering hedonic feeding (glutamatergic neurons) or the development of obesity (GABAergic neurons). This also highlights a contribution of the glutamatergic CB1 receptors to distinct aspects of the addictive-like process for food.

To further explore at which anatomical level CB1 modulates energy balance, mutant mice with a deletion of the CB1 receptors expressed in forebrain and sympathetic neurons have been analyzed (CamK-CB1KO) [135,146]. The conditional mutant mice show lower body weight than WT, but less pronounced than full KO, both on regular or high-fat food, with similar food intake, suggesting that CB1 expressed in forebrain and sympathetic neurons does not fully account for the lean phenotype observed in the complete CB1KO mice [146]. CamK-CB1KO also show reduced food intake in a fasting–refeeding paradigm [135,146]. Together, CB1 signaling in forebrain and sympathetic neurons plays an important role in body weight gain, fat accumulation, and metabolic alterations associated with chronic HFD consumption.

A conditional mutant where CB1 receptors are deleted in the dorsal raphe (TPh2-CB1KO) does not display an altered phenotype [135]. Other conditional mutants have been tested in the HT, which plays a critical role in regulating energy balance, for palatable food intake. A Hyp-CB1KO was generated using viral infection in the HT of floxed mice [157]. These mutant mice show a marked decrease of CB1 expression in the HT (60%). Their body weight is decreased on a chow diet, despite no changes of food intake, probably revealing increased energy expenditure. Interestingly, mice lacking CB1 receptors in the hypothalamic ventromedial nuclei VMN (SF1-CB1KO) display normal weight and food intake on chow [155], a reduced fasting-induced food intake [135], and surprisingly, an increase of weight and food intake on a HFD [155]. CB1 receptors in VMN neurons provide a molecular switch adapting the organism to the dietary change. Conditional mutants for the paraventricular nucleus of the HT (PVN neurons, Sim-CB1KO) show no altered phenotype on a chow diet [156] or on fasting–refeeding paradigm [135], whereas reduced body weight and reduced fat mass are observed despite similar food intake on HFD maintenance [156]. These findings reveal a diet-dependent dissociation of the role of these CB1 receptors, whose activation play a critical role under HFD exposure, potentially favoring body weight gain and obesity [156].

Conditional CB1 knockout mice in the periphery

A few conditional CB1KO mice have been used to dissect the role of CB1 receptors in peripheral systems, specifically targeting the liver, where CB1 receptors are highly expressed [138]. Mice with a hepatocyte-selective deletion of CB1 receptors (hCB1KO) display an obese phenotype on HFD but remain insulin-sensitive, suggesting the involvement of hepatic CB1 in whole-body insulin resistance [139]. In mutant mice expressing CB1 receptors only in the liver (htgCB1KO), HFD induces a small increase of hepatic triglycerides compared to WT mice, whereas no changes are observed in CB1KO [172]. In contrast, htgCB1KO mice are as resistant to insulin as WT, whereas full KO are insulin-sensitive. In another study, parallel experiments on hCB1KO and htgCB1KO mice reveal that hepatic CB1 activation is both necessary and sufficient to account for diet-induced insulin resistance, independent of body weight. Indeed, htgCB1KO mice display similar weight as WT mice on a chow diet but do not show weight gain on HFD. Interestingly, htgCB1KO mice show insulin resistance upon HFD, an effect that is not observed in full CB1KO [137], as in [172]. Together, these results confirm that hepatic CB1 receptors play an important role in insulin resistance and that they are sufficient to account for HFD-induced insulin resistance. Targeting specifically these receptors may be of strong interest for therapeutic intervention.

A study investigated the role of CB1 receptors expressed in the upper small-intestinal epithelium in the Western-diet (high fat/sugar) condition using a conditional mutant (Int-CB1KO) [159]. The Int-CB1KO mutant mice display similar weight and food intake as their WT controls. An evaluation for Western-diet preference versus a standard diet reveals that the preference of these mice is drastically reduced in the first hours of the exposure (up to 6 h over 24 h), similarly to a systemic CB1 antagonist treatment [159]. These results indicate that CB1 receptors of the intestinal epithelium are required for acute Western palatable food preferences, and highlight a gut–brain communication in food preference.

Knockout of CB2 receptors

Several studies have investigated the role of CB2 receptors in feeding behaviors, as these receptors are expressed, besides immune cells or a few brain structures, in several peripheral organs responsible for the control of metabolism. Full CB2KO male mice at 2 months display a similar weight as WT mice, whereas a significant increase (more than 25%) is observed at 12 months [160]. Similarly, food intake is increased by 40% in these mice and fat accumulation in adipose tissue is observed, indicating a gradual development of obesity on regular chow. Interestingly, similar results are described in a study investigating pain sensitivity in female CB2KO mice, with increased body weight and significantly greater food intake [162], or in male CB2KO mice that are group-housed [163]. Opposite results are described in mice overexpressing CB2 (CB2xP), with a decreased body weight from 18 weeks of age and a decreased fasting–refeeding phenotype [165]. Remarkably, when fed with HFD, male CB2KO mice show a reduced body weight gain, no difference in food intake, and insulin sensitivity [160]. In another study investigating obesity, the absence of CB2 receptors decreases body weight gain upon HFD and prevented obesity-associated inflammation, insulin resistance, and fatty liver [161]. In a more sophisticated paradigm under a short operant protocol (24 sessions of FR1), there is no difference in sucrose intake for CB2KO mice compared to WT mice, with similar breaking points [50], suggesting no acquisition or motivation deficit in these mice. A very recent study [164] investigated the phenotype of genetically modified mice either lacking (CB2KO) or overexpressing (CB2xP) CB2 receptors trained in a well-established operant food-addiction model [151]. In these conditions, mice have access to chocolate-flavored pellets during one-hour daily operant sessions over a long period of access (>100 sessions). All mice progressively increase the number of reinforcements across FR5 sessions, but CB2KO display reduced responding, and, moreover, a low percentage of these mutants develop food addiction during the early training period, suggesting a resistant phenotype. These results contrast with previously cited results, with increased food intake in CB2KO. In this previous study [160], mice are evaluated for homeostatic control of food intake (chow). In contrast, in this long-term paradigm, mice are normally fed when trained to seek chocolate pellets, indicating that the hedonic control of food intake is evaluated, which could explain this different phenotype [160]. In opposition, mice overexpressing CB2 receptors display a reduced responding only in the early period and increased responding in the late period, suggesting a switch in the reward sensitivity for these mutants with time [164]. Controversial results may arise from distinct strains of mice, different rewarding stimuli, or length of operant protocols, but, collectively, these results suggest that CB2 receptors are involved in the control of food intake and, more precisely, in processes underlying vulnerability to developing food addiction.

Altogether, genetically modified mice combined with sophisticated models of addictive-like eating behavior bring new insights in deciphering more precisely the role of the distinct components of the ECS in vulnerability to food addiction.

## 4. Discussion on the Role of the ECS in BED

### 4.1. Peripheral Endocannabinoid Regulations in BED

In patients with BED, blood levels of AEA are differently regulated depending on food preferences. Indeed, levels decrease after consuming a non-preferred food, increase after consuming a preferred food, and are correlated to the subject’s sensation of the urge to eat [117]. These observations are consistent with preclinical data showing the impact of peripheral endocannabinoid administration on increased palatable food intake [173,174] and reward system activation [175]. In rodents, increasing the endocannabinoid tone by intravenous administration of exogenous AEA and 2-AG enhance DA transmission through a CB1 receptor mechanism [176,177]. As proposed for other eating disorders [178], the increased endocannabinoid tone observed in patients suffering from BED (AEA and 2-AG) and animal models (AEA) (Figure 2B) could be a consequence of repeated palatable food access and might contribute to facilitating the rewarding and motivational properties of palatable food, leading to aberrant feeding behaviors. Chronically elevated AEA levels could also contribute to an enhanced DA release in the mesocorticolimbic system, leading to a progressive desensitization of the DA system that could contribute to the “binge-eating vicious cycle” (Figure 1A).

### 4.2. Central ECS Regulations and Brain Structures Involved in BED

#### 4.2.1. Central Regulations

Centrally, the ECS appears to be globally downregulated in the reward system in patients suffering from BED regarding endocannabinoid levels, receptors, and enzymes. CB1 receptors appear to be a crucial element in the mediation of altered palatable food reinforcing and motivational properties observed in binge eating. Importantly, the final effects of ECS alterations on DA activity highly depend on the functional balance between GABA and glutamate inputs to local neuronal populations, as confirmed by genetically modified models (see Table 1).

#### 4.2.2. PFC-Striatal Connections

Most of the literature describes NAc connectivity with PFC in the context of BED. Activation of CB1 receptors decreases excitatory glutamatergic transmission in the VTA and the NAc, mainly by regulating the activity of neurons projecting from the PFC [179]. Direct CB1 agonism leads to rewarding effects in animals, and antagonists reduce the reinforcing effects of palatable food and drugs of abuse. Cannabis increases DA levels in the ventral and dorsal striatum and is associated with the subjective value of reward [180]. Moreover, both DA and endocannabinoids are involved in reward-driven palatable food intake as their levels correlate with the craving for palatable food [181,182]. In the NAc, local injections of AEA enhance the reward associated with palatable food [183] and local injections of 2-AG increases feeding even in sated rodents [33]. Thus, a decrease of AEA in NAc, and CB1 mRNA and density in PFC and NAc could be a compensatory mechanism to invert the exacerbated palatable-food-associated reward and motivation in binge eating. Moreover, a decrease of presynaptic CB1 receptors and endocannabinoids at glutamate terminals in the NAc and PFC could be an attempt to restore the cortico-cortical and cortico-striatal connectivity underlying the altered impulse control over palatable food consumption (Figure 2C). As animals had intermittent access, we cannot exclude the possibility that changes in endocannabinoid levels could be a consequence of fasting more than binge-eating behavior. However, it has been shown that fasting leads to the opposite effect with increased limbic levels of AEA and 2-AG that return to normal after refeeding [33].

#### 4.2.3. HPC and Amygdala

Decreased endocannabinoid levels have also been reported in the HPC and the amygdala following binge eating. Conflictive findings have been obtained concerning the role of the endocannabinoids in the HPC, with intra-hippocampal injections of cannabinoid agonists impairing [184,185] or enhancing [186] memory encoding or consolidation. Interestingly, the local blockade of CB1 receptors impairs the consolidation of inhibitory avoidance training [187]. Data concerning the effects of endocannabinoids in the amygdala appear to be more consistent. Injection of the inverse CB1 agonist/antagonist AM251 in the amygdala complex impaired the memory reconsolidation of fear learning [188]. On the other hand, activation of CB1 receptors in the basolateral amygdala enhances memory reconsolidation and is important for glucocorticoid effects on memory consolidation [189]. According to extensive evidence about the roles of HPC and amygdala in memory consolidation and emotionally arousing experiences [190], a reduced local endocannabinoid tone in BED could act as a compensatory mechanism to reduce palatable-food-related emotional memory, including the stress component in binge eating. 

#### 4.2.4. Hypothalamus

In the HT, CB1 receptor transcripts and AEA levels are oppositely regulated following binge eating compared to what is observed in reward-related brain regions. Hypothalamic CB1 receptors are highly involved in energy balance [157]. Intra-hypothalamic infusions of CB1 receptor agonists (AEA) increase food intake [173,174] and CB1 blockade in the PVN increases fasting-induced hyperphagia [191]. This is in contradiction with PVN conditional KO data, where no phenotype change is observed in this same paradigm [135]. This may be due to compensatory mechanisms in mutant mice or to the high-functional heterogeneity of the HT subnuclei. In the HT, enhanced levels of AEA reflect what is globally observed at a plasmatic level and could be more linked to the feeding status of animals.

## 5. Conclusions

In summary, studies in animal models and patients with BED demonstrate that dysregulation of the ECS physiology can have detrimental or protective effects on eating behaviors through changes in mesocorticolimbic function, and contribute to the pathogenesis and maintenance of the disorder. Future research is needed to draw a more complete picture of the brain and peripheral ECS configuration in binge eating and whether it is altered depending on macronutrient profile, gender, and access schedules in animal models. Additionally, ECS alteration in BED leads to a chicken-and-egg question on whether these alterations are a consequence of the disorder or serve as risk factors.

Strategies for treatment are not successful with classical cannabinoid ligands due to the complexity of the circuits involved in ECS activity, and due to potential emotional side effects. Compounds that would allow regulation of the endogenous tone by acting on the synthesis or degradation of endocannabinoids or indirectly acting on receptor activity with biased cannabinoid modulators may be more appropriate to specifically target BED [43,192]. Novel molecular approaches have been proposed for treating feeding behaviors [13,193] but interdisciplinary strategies combining psychotherapies with pharmacology still appear to be the best options to treat patients suffering from BED [62]. Fine dissection of the ECS neuronal circuitry involved in binge eating using innovative optogenetic, chemogenetic, and fiber-imaging tools will provide better knowledge on how to specifically target treatments for this disorder. As BED involves complex underlying mechanisms with altered reward-related and impulsive behaviors and impairments of ECS functions, such new treatment options to normalize these behaviors without adverse side effects would be beneficial for other disorders involving addictive behaviors.

## Figures and Tables

**Figure 1 ijms-24-09574-f001:**
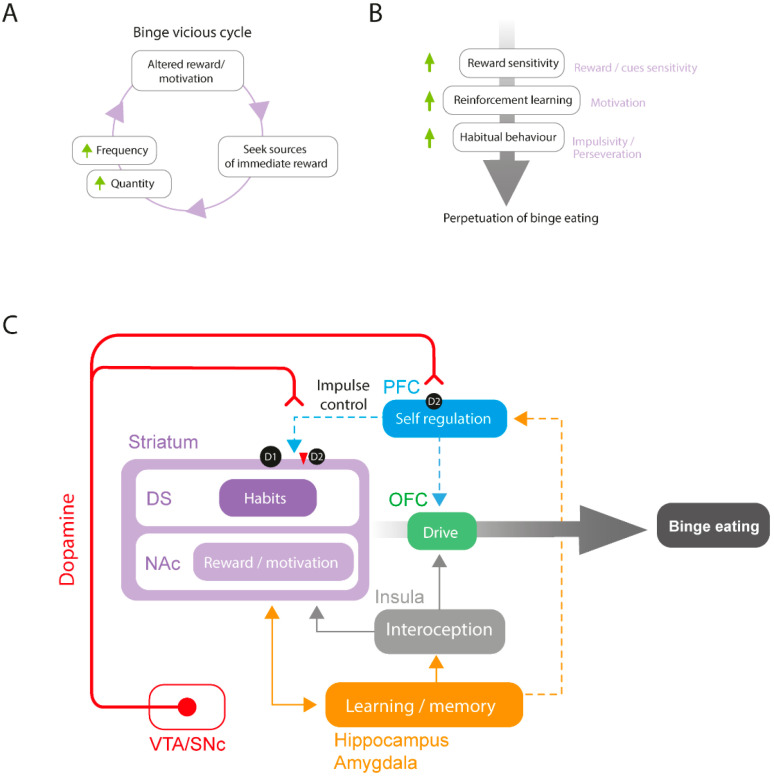
Neurobiology of binge eating: (**A**) Binge-eating vicious cycle. (**B**) Altered reward processes in binge eating. (**C**) Proposed model of brain-connectivity alterations in binge eating disorder. This model focuses on DA activity, and other neurotransmitter systems (opioid, serotonin, and cannabinoid) are not represented.

**Figure 2 ijms-24-09574-f002:**
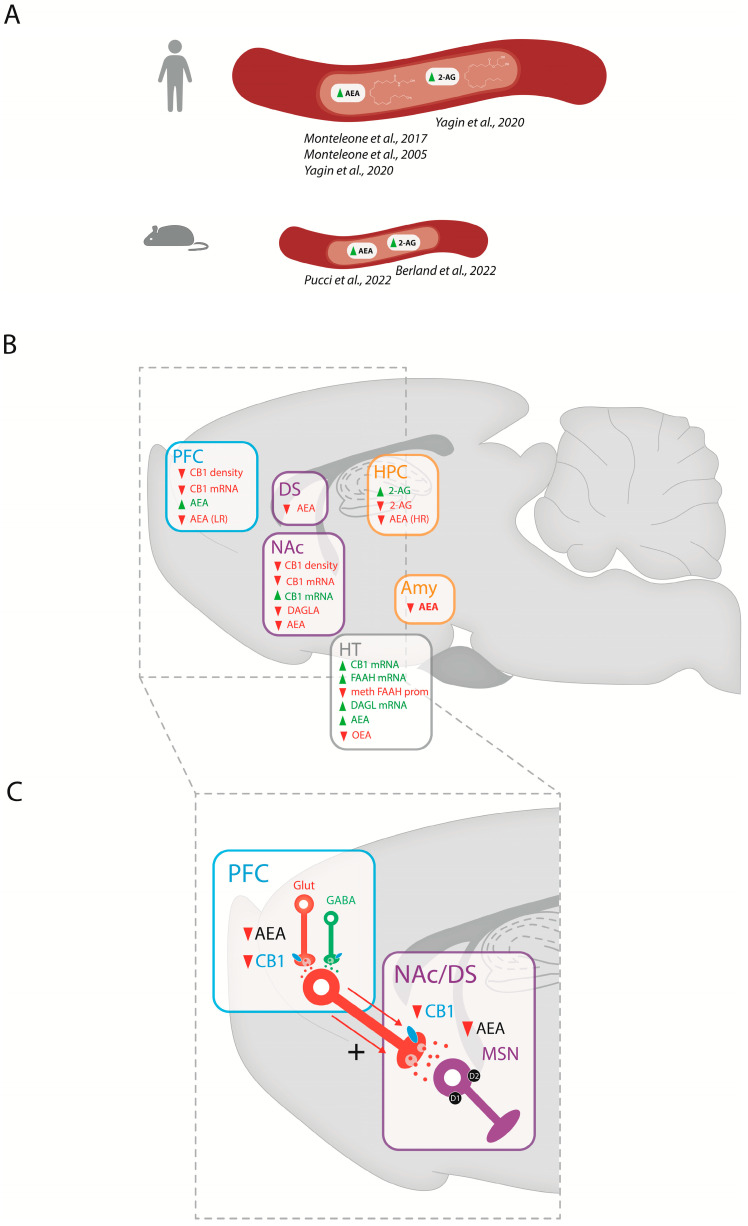
Central and plasmatic alterations of the endocannabinoid system in BED: (**A**) Peripheral ECS adaptations in BED in humans and animal models. (**B**) Central ECS adaptations in BED preclinical models. (**C**) Proposed model of ECS activity in PFC-striatal connections. Literature shows decreased connectivity between PFC-NAc as well as cortico-cortical connectivity. Animal models of BED show decreased endocannabinoids and CB1 in glutamate terminals of NAc and PFC. Activation of CB1, as it is coupled to Gi, inhibits neurotransmitter release from presynaptic terminals. In BED conditions, there would therefore be a decreased inhibition of glutamate release, both from glutamatergic presynaptic terminals in PFC and NAc. It could also impact GABA transmission in PFC (presynaptic GABA, in green) and in NAc (postsynaptic MSN in purple). This could be a balanced mechanism to invert the exacerbated palatable-food-associated reward and motivation in binge eating [116,117,118,119,120].

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
