# Peer review of "The Role of the Endocannabinoid System in Binge Eating Disorder"

_ijms, 2023, doi:10.3390/ijms24119574_

Round 1

Reviewer 1 Report

When introducing the role of the ECS in food intake, I would omit the first two sentences that refer to emotional processes, pain management or drugs (page 2), and focus only on the food intake. The increase in appetite or "munchies" (appetite stimulation, especially for sweet and savory foods) is probably the most well-known effect of THC. The Authors could have made a better analogy to the scope of this review by discussing this phenomenon. One of the publications worth considering is Levichev and colleagues (Current Biology 2023, 33: 1–15).

Quality of English is generally OK. I encourage Authors to simply/shorten their text. It will read and flow much better

Author Response

"Please see the attachment

Reviewer 2 Report

Romain Bourdy and Katia Befort wrote a review with the purpose of updating current knowledge on the neurobiology of BED and to highlight the specific role of the endocannabinoid system (ECS) in its development and maintenance. The authors proposed a model for a better understanding of the mechanisms involving the endocannabinoid system and deepen animal and human studies on this topic. 

I want first to congratulate with the authors for their big effort in putting together a lot of preclinical and clinical results on this interesting topic. The overall quality is good. Although the work is scientifically sound, I found difficult in reading it and I felt confused on what really was the main aim of the work. Sometimes authors refer to binge eating behavior not only for binge eating disorder (BED) but also for other eating disorders (i.e., bulimia nervosa). I was wondering if it might not be useful to consider binge eating in a transversal way as a binge spectrum and not as a diagnosis and use the “binge eating behavior” as main objective of this review.

In current form, the title is misleading! The reader expects to read about the endocannabinoid system but, after the first paragraphs where there is a description of the system, evidence on the ECS is elucidated after more than half of the review. Indeed, the review begins with a precise and complete examination of the dopaminergic system and the ECS is explained only halfway through. This first part appears redundant and not useful for the main aim of this review, creating a lot of confusion in the reader.  I wondered if it might not be better to reduce this section very much to a small paragraph and focus on the ECS, as I expect to read considering the title. Alternatively, it would be better to change the focus of the review talking about both systems changing the focus and the title. Furthermore, the authors only examine the reward system and no other systems implicated in the neurobiological of the BED. I think that the goal is too ambitious and that perhaps it could be useful to divide the contents into two works: one for updating the neurobiology of BED (or binge eating spectrum); the other one on the role of the endocannabinoid system in BED (or binge eating behavior).

Here are some major comments:

Title 

Why “Binge eating” and not “binge eating disorder” is used in the title? Considering the aim of this review (on BED), it would be useful to change the title as “The role of the endocannabinoid system in binge eating disorder”. On the other hand, reading the whole manuscript, I had the feeling that authors used binge eating, binge eating disorder and binge eating behavior in an interchangeable way. In this light, it might be useful to consider the binge spectrum and use the “binge eating behavior” through this review.

Abstract 

First sentence can be deleted. The phrase “BED modulates the brain reward circuit both in humans and animal models, which involves dynamic regulations of the dopamine circuitry. The endocannabinoid system plays a major role in food intake regulation, both centrally and in the periphery” should be rephrased. It is not clear. First, BED does not modulate brain reward circuit!!! The alteration in reward system and dopamine regulation is considered one of the contributing factors in the development and maintenance of BED both in humans and animal models. Alongside this, the endocannabinoid system also plays a major role in food intake regulation, both centrally and in the periphery. Second, if the aim moves to binge eating behavior, the reward system is one of the pathways involved in the onset and maintenance of this and other addictive-like eating behaviors. 

Paragraph 1.2.1. Central role of food intake. 

I want to highlight that not only central but also peripheral pathways control food intake. I would suggest using only “food intake control” without “central”.

At the end of this paragraph “Some of these paradigms will be described throughout this review to illustrate specific studies investigating the role of the ECS in food intake and more specifically in binge eating.” Add “binge eating behavior (or binge eating disorder). 

Paragraph 1.2.2.  Role of the endocannabinoid system in food intake 

The authors stated that “For example, cannabis is well known for activating hunger by stimulating the appetite for sweet substances”. The authors should be aware that hunger and appetite refer to different meaning. In the paper cited (reference: Cannabis: effects on hunger and thirst), it is explained the effect of cannabis on hunger stimuli, not appetite. Please rephrase the sentence not using appetite word.  

At the end of the 1.2.2. authors stated: “Whether this system is altered in eating disorders and associated pathologies needs to be clearly examined for a better understanding of the adaptations of the hedonic balance in such situations”. They refer to all eating disorder, not to BED.  Please, clarify. 

Paragraph 1.2. The endocannabinoid system regulates food intake

I think that you can avoid using two paragraphs. I would suggest reducing the paragraph 1.2.1. and merge it with the paragraph 1.2.2.  The focus of the work is not food intake control but the ECS, isn’t it? 

Paragraph 2. Eating disorders and obesity 

Why do you mention “obesity” if it is not detailed and explained in the paragraph? If the focus of the review is BED or binge eating behavior, delete "and obesity" and change the title to "Binge Eating Disorder" or “Binge eating behavior” or “Binge eating spectrum”. 

Please, delete 2.1. A serious health threat. You should use one single paragraph to briefly describe BED or Bing eating spectrum disorders or binge eating behavior (in eating disorders). 

The sentence: “Eating disorders are psychiatric conditions listed in the Diagnostic and Statistical Manual of Mental Disorders Version 5 (DSM-5) [52], and include anorexia nervosa (AN), bulimia nervosa (BN), and binge eating disorder (BED) [53]. They are characterized by maladaptive food patterns, and may in some cases be associated with compensatory purging behaviors” is not useful and unnecessary. 

In this paragraph the authors stated. “Loss of control towards food intake also occurs in other eating disorders [58] and is a likely contributor to the growing obesity epidemic [59-61].” This is not completely true: (1) although loss-of-control related behaviors are frequently reported in other eating disorders such as anorexia or bulimia, it is NOT a likely contributor to the growing obesity epidemic in patients suffering from anorexia or bulimia!!! It could be true only for BED. 

Then, the authors stated: “Indeed, more than 70% of individuals suffering from eating disorders report physic and psychosocial comorbid disorders, including obesity, anxiety, and substance use.” First, it is important to mention that the meaning of the sentence is not clear. Second, the reference is missing! Third, I wonder why people suffering from eating disorders (any eating disorders???) should report obesity among physical comorbid disorders???This is NOT true for anorexia or bulimia!!! If you want to use this sentence, you must better reformulate it and add reference. I would suggest rephrasing the whole paragraph focusing only on BED or binge spectrum or binge eating behavior. Furthermore, I suggest deleting Figure 1A, it is not informative, and it is not the focus of the review. 

2.2. Neurobiology of binge eating disorder 

Delete the subparagraph. You can merge this paragraph with the previous one and use the title “Epidemiology and neurobiology of binge eating disorder” or simply “Neurobiology of binge eating disorder” and use “binge eating disorder” or “binge eating spectrum” or “binge eating behavior” accordingly to the aim of the work. 

In this part of the paragraph, you should also mention not only dopaminergic pathway but also hormones or neuroactive peptides such as sex hormones or gut hormones (e.g., leptin, ghrelin, nesfatin) that regulate body homeostasis and that are often altered in binge eating disorders disturbing normal food reward circuits and the opioid pathway that plays an important role in feeding behavior driving the hedonic experiences of food intake. 

At the end of the paragraph “and open new avenues for treatment strategies for people with BED or obesity”: again “obesity” is to be deleted. Obesity is not an eating disorder and, in this review, it is not the focus. 

Paragraph 2.2.1. Neurobiology of BED (Human studies)

Delete “Neurobiology of BED”. You should use only “Human studies”. 

Please rephrase “In eating disordered patients that binge eat”. It sounds so bad. You should say “in patients reporting altered eating behaviors such as binge eating”. 

Paragraph 2.2.2. Neurobiology of BED (Animal models)

Delete “Neurobiology of BED”. You should use “Animal models”. 

Reference is missing here: “DA plays a crucial role in various functions such as food craving, decision-making, executive functioning, and impulsivity traits, factors that collectively contribute to the on- set and persistence of binge eating as observed in BED patients”.

Paragraph 3. Binge eating disorder and the endocannabinoid system 

I think this is the focus of the review and that paragraph 2 is redundant, excessive, and not helpful as well as very long. 

Again: “Interestingly, studies in patients with eating disorders and obesity have revealed relationships between ECS gene polymorphisms …” eating disorders in general or BED???? 

Paragraph 3.1. Endocannabinoid levels in BED 

In BED, binge spectrum or in binge eating behaviors? The authors mentioned both BED and other eating disorders and also obesity.  

Authors stated “Indeed, blood levels of AEA, but not 2-AG, are increased in both AN and BED drug-free women, an effect that is not observed in patients with BN.”  Why AN? You should add more information. The reader could be confused by this sentence. 

This paragraph investigated animal research and clinical studies. I would suggest dividing it into two subparagraphs: “Animal models” and “human studies”. You should use this suggestion for other paragraphs that investigate both animal and human studies. 

3.3. Cannabinoid receptors in BED

The authors intend BED or Binge spectrum, or binge eating behavior? I suppose, binge eating behavior. 

3.3.1. Transcriptional regulations of cannabinoid receptors 

Reference is missing: “Changes in endocannabinoid signaling, largely mediated by CB1 receptors, are linked to the development of binge eating disorder.”

3.3.2. Genetic animal models targeting the cannabinoid receptors to study food intake and addictive-like eating behaviors

In the paragraph and subparagraphs, the authors reported studies and preclinical results associated with various eating disorders, as well as food addiction and obesity. I would suggest including more background in the introduction on food addiction (as addictive-like eating behaviors often reported in patients with BED, BN, AN, in binge eating spectrum and obesity) and obesity.   

The Proposed model of ECS activity in PFC-striatal connections is mentioned but not clearly explained in the text. Please, use a paragraph to better explain or reassume the mechanisms involving the endocannabinoid system and your proposed model. 

Other minor comments:

Please, use the first-person language. Please, not use “individuals with BED” or “BED patients” or “obese patients”. The correct mode is using “patients with BED” or “patients suffering from BED” or “people with/suffering from BED”. Also “obese patients with BED” is not correct. You should say “patients with obesity and BED”.

Please, carefully check the acronyms through the manuscript. 

Figure 2C: please provide in the legend a briefly explanation of the proposed model of ECS activity in PFC-striatal connections.

Check carefully English language throughout the manuscript (e.g., Acknowledgments not “Aknowledgments”). 

Author Response

"Please see the attachment

Round 2

Reviewer 2 Report

The authors reviewed the manuscript according to all my suggestions. Only minor revision is required. 

Please, use the first-person language. It is very important to use it to maintain an environment of dignity, respect and hope. Placing the person first and the disability second helps eliminate stereotypes that can form. 

Please change:

- “individuals with BED” (in 2.2. Human studies), 

- "BED individuals" (in 4.2.1. Central regulations) 

- "obese patients" (in 2.2. Human studies, in 3. Binge eating disorder and the endocannabinoid system, and in 3.1. Endocannabinoid levels in BED)

in “patients with BED/obesity” or “patients suffering from BED/obesity” or “people with/suffering from BED/obesity”.  

Thank you.

Minor editing. 
